# Upconversion Emission Studies in Er^3+/^Yb^3+^ Doped/Co-Doped NaGdF_4_ Phosphor Particles for Intense Cathodoluminescence and Wide Temperature-Sensing Applications

**DOI:** 10.3390/ma15196563

**Published:** 2022-09-21

**Authors:** Abhishek Kumar, Helena Couto, Joaquim C. G. Esteves da Silva

**Affiliations:** 1Chemistry Research Unit (CIQUP), Institute of Molecular Sciences (IMS), Departamento de Geociências, Ambiente e Ordenamento do Território, Faculdade de Ciências, Universidade do Porto, Rua do Campo Alegre s/n, 4169-007 Porto, Portugal; 2Pranveer Singh Institute of Technology (PSIT), Kanpur-Agra-Delhi National Highway (NH-19), Bhauti, Kanpur 209305, India; 3Instituto de Ciências da Terra—Pólo Porto, Departamento de Geociências, Ambiente e Ordenamento do Território, Faculdade de Ciências, Universidade do Porto, Rua do Campo Alegre s/n, 4169-007 Porto, Portugal

**Keywords:** NaGdF_4_:Er^3+^/Yb^3+^, upconversion, luminescence, cathodoluminescence, temperature sensor

## Abstract

Er^3+^/Yb^3+^ doped/co-doped NaGdF_4_ upconversion phosphor nanoparticles were synthesized via the thermal decomposition route of synthesis. The α-phase crystal structure and nanostructure of these particles were confirmed using XRD and FE-SEM analysis. In the power-dependent upconversion analysis, different emission bands at 520 nm, 540 nm, and 655 nm were obtained. The sample was also examined for cathodoluminescence (CL) analysis at different filament currents of an electron beam. Through CL analysis, different emission bands of 526 nm, 550 nm, 664 nm, and 848 nm were obtained. The suitability of the present sample for temperature-sensing applications at a wide range of temperatures, from room temperature to 1173 K, was successfully demonstrated.

## 1. Introduction

Rare-earth-doped phosphor nanoparticles are very useful in different kinds of luminescence, e.g., cathodoluminescence (CL), upconversion (UC), and downconversion (DC) luminescence [1,2,3]. With a broad range of luminescence properties, these rare-earth-doped phosphor nanoparticles have a broad range of applications [4,5,6,7,8,9,10,11,12,13]. To achieve effective application, the luminescence efficiency of these nanoparticles should be very high [14]. Intense luminescence can be achieved through different approaches. The main approaches are the synthesis process, doping element concentration combinations, and co-doping of different metal and non-metal elements [15,16,17,18,19,20]. Out of the above mentioned approaches, the adoption of a suitable synthesis route is very desirable [21]. The controlled particle shape/size, colloidal stability, geometrical structure of particles etc. are a result of the adoption of a proper synthesis route. There are a number of synthesis processes that have been developed in recent times. The solid-state synthesis method, chemical co-precipitation route, hydrothermal method, and thermal decomposition method are very common techniques for the synthesis of these particles [21]. The particle shape, size, morphology, crystal structure, and optical/luminescence properties are dependent on these synthesis routes [22]. For the achievement of a stable and suitable nanoparticle for a wide range of temperature-sensing and cathodoluminescence (CL) applications, the thermal decomposition route of synthesis has been adopted [21].

Temperature sensing is an important application of rare-earth-doped phosphor nanoparticles [14]. The rare-earth-doped upconversion phosphor particles have potential uses in the development of non-contact type temperature sensors [23]. To date, some temperature sensors with a limited range of temperatures and low values of sensor sensitivity have been developed with different rare-earth-doped phosphor materials. The sensing range of temperatures and sensor sensitivities of Er^3+^/Yb^3+^ dopant combinations in different phosphor hosts are summarized in Table 1. Through these comparisons it is found that the sensing ability in a wide range of temperature variation has not been achieved very well so far. The common temperature ranges from room temperature to 600 K are observed. Therefore, the wide range of sensing ability is a current requirement to make this system suitable for a variety of applications.

The developed Er^3+^/Yb^3+^ doped/co-doped NaGdF_4_ systems show cathodoluminescence (CL) ability along with temperature-sensing applications. The multifunctional ability of a material is very desirable. If a single material has a variety of applications, it will be in more demand for industrial-scale production. Therefore, this system is useful for both non-contact types of temperature sensors and display device fabrications as well. This shows the multi functionality of this system.

In the present work, Er^3+^/Yb^3+^ doped/co-doped NaGdF_4_ upconversion nanoparticles were synthesized via the thermal decomposition method. Nanosized particles with a pill shape were formed. Several structural and optical properties of these particles were examined. The XRD analysis confirmed the α-phase crystal structure of NaGdF_4_: Er^3+^/Yb^3+^ nanoparticles. Through power-dependent upconversion luminescence analysis using 980 nm continuous wave laser excitation, different emission bands were obtained. These particles can be used for temperature-sensing and cathodoluminescence (CL) applications.

## 2. Experimental Process

### 2.1. Chemicals and Method

Gadolinium oxide (Gd_2_O_3_, 99.9%), ytterbium oxide (Yb_2_O_3_, 99.9%) and erbium oxide (Er_2_O_3_, 99.9%), were used, along with hydrochloric acid (HCl, 37% concentrated), oleic acid (C_18_H_54_O_2_, 60–88%), oleylamine (C_18_H_37_N, >95%), ethanol and n-hexane. The entire synthesis process was performed in the presence of nitrogen gas.

#### 2.1.1. Synthesis Process

The synthesis process was adopted from our previous report [22]. Gd_2_O_3_, Er_2_O_3_ and Yb_2_O_3_ were used in the molar percentage ratio of 80 mol %, 2 mol %, and 18 mol %, respectively. These molar percentages of the three rare-earths were selected as, in previous research, 2 mol %/18 mol % was an optimized combination of Er^3+^/Yb^3+^ in different fluoride hosts [22]. All rare-earth oxides were dissolved in 37% concentrated HCl and heated at 60 °C until transparent solutions of hexahydrated GdCl_3_, YbCl_3_ and ErCl_3_ were obtained. The reaction process is explained as follows:2(R.E.)_2_O_3_ + 12HCl → 4(R.E.)Cl_3_·6H_2_O;
where R.E. stands for Gd, Yb and Er.

Next, in a 100 mL round bottom 3-neck flask, 10 mL of oleic acid (OA) and 15 mL of olamine (OM) along with the above hexahydrate rare-earth chlorides solution were combined. The entire system was stirred at 1000 rpm and at 140 °C constant temperature for 60 min under the constant flow of nitrogen gas to obtain a homogeneous mixture. Then, the temperature was reduced to 55 °C and a solution of NaOH (100 mg) and NH_4_F (150 mg) in methanol was injected through a syringe into the solution. To remove methanol from the mixture, the solution temperature was increased up to 70 °C under stirring for 20 min. Then, the temperature was slowly increased up to 300 °C. After one and half hours of reaction at this temperature, the thermal decomposition process of the synthesized upconversion nanoparticles was completed, and the temperature was cooled down to room temperature while stirring the synthesized nanoparticles. At room temperature, the precipitate was formed by adding an excess of ethanol and was collected by using centrifugation @ 8000 RPM, and then synthesized particles were washed 3 times with ethanol and then dried in a vacuum chamber at 60 °C overnight to obtain fine powder nanoparticles of NaGdF_4_: Er^3+^/Yb^3+^.

#### 2.1.2. Characterization Tools

XRD analysis was carried out using a Bruker D8 advanced instrument in the range of diffraction angle 25° to 75°. The surface morphology of the sample was studied using field emission scanning electron microscopy (FESEM) on a Supra 55, Carl Zeiss. The UC spectra were monitored using a CCD spectrometer (Model: ULS2048 × 64, Avantes, 2586 Trailridge Dr E, Suite 100, Lafayette, CO 80026, USA) with a 980 nm diode laser as an excitation source. Cathodoluminescence was measured on a Lumic HC3-LM instrument coupled to an optical microscope (Institute of Earth Sciences, University of Porto). The system was operated at 14 kV and a filament current of 0.18 mA. Cathodoluminescence images were acquired with a digital video camera (KAPPA PS 40C-285 (DX)) with dual-stage Peltier cooling. The temperature sensing analysis was carried out on a homemade Nano heater system.

## 3. Results and Discussion

### 3.1. Structural Analysis

#### 3.1.1. XRD Analysis

The XRD analysis was carried out in the diffraction angle range of 25° to 75°. Diffraction peaks corresponding to α-phase of the NaGdF_4_ crystal structure were observed and all peaks were well matched with the standard data file JCPDS No 27-0697, as observed and shown in Figure 1 [23]. Some small peaks were also observed, along with dominant peaks which mostly had intensities equivalent to the background. These small peaks may arise due to impurities from synthesizing agents or possible formation of extra phases of the present sample. These peaks may be ignored as they can be easily removed in baseline correction. A comparatively intense peak around 43° was seen. This small peak was probably due to some impurities from synthesizing agents.

#### 3.1.2. Nanostructure and Elemental Analysis

The nanostructures of the synthesized upconversion nanoparticles are shown in Figure 2. The FE-SEM image was taken using a 20 nm scale bar. Mostly pill-shaped but nearly spherical particles were observed (Figure 2). The particle morphology was not very regular but all particles were well distinguished. There was no agglomeration effect observed. These particles may also be stable in colloidal states through some surface modification using chemical procedures. Therefore, these particles are very suitable for multifunctional applications in a broad range of fields.

### 3.2. Optical Characterization

#### 3.2.1. Upconversion Spectra Analysis

The power-dependent upconversion emission spectra from the present sample NaGdF_4_: Er^3+^/Yb^3+^ were recorded using 980 nm CW diode laser excitation [33]. The pump power of excitation was varied from 100 mW to 2600 mW with an interval of 100 mW and 200 mW as shown in Figure 3. Here, in the range of 450–700 nm of visible emissions, different emission bands at 490 nm, 520 nm, 540 nm, and 655 nm were obtained. These bands had energy level transitions at ^4^F_7/2_→^4^I_15/2_, ^2^H_11/2_→^4^I_15/2_, ^2^S_3/2_→^4^I_15/2_, and ^4^F_9/2_→^4^I_15/2_, respectively. At a lower excitation power of 100 mW, the upconversion emission intensity was very low. With the increase in pump power of excitation, the emission intensity was increased correspondingly up to 2600 mW of excitation. Due to the limit of the pump power of the excitation source, no power beyond 2600 mW was selected, but it is predicted that further enhancement in emission intensity is possible with higher excitation power selection. The relative emission intensity variation of different emission bands was also observed. For example, at low excitation power up to 800 mW, corresponding to the ^2^H_11/2_→^4^I_15/2_ emission band, the intensity was lower than the emission intensity of the ^2^S_3/2_→^4^I_15/2_ emission band, but beyond 800 mW this emission intensity was relatively inverted. This probably happened due to excitation power-dependent population differences from different emission bands. A detailed description of the pump power of excitation-dependent emission intensity variations is given in the following sections.

In Figure 4a, the variations of emission intensities with the pump power of excitations for different dominant emission bands at 520 nm, 540 nm, and 655 nm are summarized. In this graph, the emission intensities for thermally coupled bands at 520 nm and 540 nm increase constantly with the increase in pump power. Corresponding to 655 nm, the emission intensity first increased to 800 mW and beyond this power the emission band was almost constant, up to 2200 mW. The further increase in power for this band showed little change in emission intensity. In Figure 4b, the variation of intensity ratio between two thermally coupled levels at 520 nm and 540 nm with the pump power of excitation is shown. This graph explains the relative variation of intensity between these levels. From 100 mW to 400 mW, this ratio decreased, then beyond this power this ratio increased. In the same manner, in Figure 4c the ratio of total green (520 nm + 540 nm) and total red (655 nm) variation with the pump power of excitation is shown. At 100 mW, this ratio value was higher, but beyond this power, the ratio varied linearly from lower to higher values.

The Commission Internationale de l’Elcairage (CIE) plot is an important tool to examine the purity of emitted luminescence [34]. The corresponding CIE plot for the pump power of excitation-dependent intensity variation of the present sample is shown in Figure 4d. Here, with the increase in pump power of excitation, the color coordinates and the color purities both shift to the green region from the central region. This variation shows that the radiative emission probability was increased with an increase in excitation power. In Appendix A, the pump power of excitation-dependent CIE coordinates and the corresponding color purity are summarized. Corresponding to the lowest excitation at 100 mW, the purity of the green color emission was only 2.0%; with the increase in excitation power, the emission color purity was also increased, and it achieved a maximum of 54.0%, corresponding to 2200 mW. This confirms that the radiative emission efficiency increased with an increase in excitation power.

#### 3.2.2. Cathodoluminescence Analysis

Cathodoluminescence (CL) is a well-known luminescence process with a broad range of applications in display devices [35]. Here, the emission spectra of NaGdF_4_: Er^3+^/Yb^3+^ upconversion phosphor particles with the variation of the filament current of cathode rays are plotted in Figure 5. From 0.10 mA, the value of the filament current to 0.70 mA is selected. Here it is observed that with the increase in this parameter, CL emission intensity was also increased. This increase in CL emission intensity is the same as the power dependence upconversion emission spectra described in the previous section. In comparison to UC spectra, the emissions from 520 nm and 540 nm emission states are not completely distinguished. It seems there is a broad CL emission in the range of 513 nm to 573 nm from these states. The emission from the ^4^F_9/2_ level also shifted from 655 nm of UC to 664 nm. These shifts in emission bands are probably due to the emission probabilities selections due to differences in the selection of excitation sources. In the CL spectra, the emission band corresponding to ^4^F_7/2_→^4^I_15/2_ (490 nm) did not appear, but an additional emission band at 848 nm emission through ^4^I_9/2_→^4^I_15/2_ was obtained.

A camera image under white light illumination of the prepared sample for cathodoluminescence analysis is shown in Figure 6a. In a dark environment, CL images at different filament currents were taken and are summarized in Figure 6b–j. In these images, the brightness increased with an increase in filament currents at 0.10 mA, 0.20 mA, 0.30 mA, 0.40 mA, 0.50 mA, 0.60 mA and 0.70 mA. The increase in filament current ensures the availability of more electrons within the boundary of the fabricated sample. This increased number of available electrons is responsible for the improvement in the CL brightness and clarity of the taken photographs. This property of the present sample is very interesting and confirms that this sample is suitable for display applications. The corresponding CL emission spectra of these CL photos are described in the next section.

#### 3.2.3. Energy Level Transitions for Both Luminescence

In Figure 7, a combined energy level diagram for upconversion emission bands and CL emission bands is shown. For UC emissions, the energy transfer (ET), excitation state absorption (ESA) and ground-state absorption (GSA) processes took place [33]. Furthermore, in the CL process, the incident electron beams are directly absorbed by activator ion Er^3+^ and are excited to higher states. From there, some radiative transitions took place along with non-radiative transitions. We thus tried to explain the CL process through an energy level diagram for the first time; as electron beams are energetic particles, they knock the grounded particles to an excited level, and then the luminescence process can take place.

## 4. Temperature-Sensing Application of the Present System

For the temperature-sensing study, a pellet was made from the powder sample using a pelletizer. This pellet was then placed inside a homemade chamber with controlled heating. Through 980 nm diode laser excitation, the upconversion luminescence was recorded on a CCD spectrometer. The temperature of the sample varied from 301 K to 1173 K with an interval of 50 K. The fluorescence intensity ratio (FIR) was calculated for thermally coupled 520 nm (^2^H_11/2_→^4^I_15/2_) and 540 nm (^4^S_3/2_→^4^I_15/2_) levels as per [14]:(1)$$FIR=\frac{I_{520}}{I_{540}}=\frac{W_{H}g_{H}\nu_{H}}{W_{S}g_{S}\nu_{S}}exp\left(\frac{-\Delta E}{k_{B}T}\right)=Bexp\left(\frac{-\Delta E}{k_{B}T}\right)$$
where I_520_ and I_540_ are the integrated intensities corresponding to the ^2^H_11/2_→^4^I_15/2_ and ^4^S_3/2_→^4^I_15/2_ transitions, respectively. *W_H_* and *W_S_* are the radiative transition probabilities, g_H_ and g_S_ are the (2J + 1) degeneracy of levels ^2^H_11/2_ and ^4^S_3/2_, respectively, and ν_H_ and ν_S_ are the photon frequencies of the ^2^H_11/2_ →^4^I_15/2_ and ^4^S_3/2_ →^4^I_15/2_ transitions, respectively. ΔE is the energy gap between the two emitting levels, k_B_ is the Boltzmann constant and T is the absolute temperature. The above equation can be written as:(2)$$\ln\left(FIR\right)=\ln B+\left(-\frac{\Delta E}{k_{B}T}\right)$$

To obtain the value of ΔE, the plot between ln FIR and inverse temperature (1/T) is shown in Figure 8a and the slope (−535.0) of this plot gives the value of ΔE/k_B_. The sensing ability of this sensor is confirmed through the measurement of its sensitivity. The absolute and relative sensitivities are calculated by [36]:(3)$$S_{A}=\frac{d(FIR)}{dT}=R\frac{\Delta E}{k_{B}T^{2}}$$
(4)$$S_{R}=\frac{1}{R}\frac{d(FIR)}{dT}=\frac{\Delta E}{k_{B}T^{2}}$$

In the Figure 8c the absolute sensitivity (S_A_) is measured using formula (3). The relative sensitivity for this sensor was plotted in Figure 8d. This result shows that around room temperature, the sensor sensitivity is the highest. The measurements of sensitivity against temperature variation prove that the temperature sensor based on this material has a good sensitivity over a large temperature variation. The absolute sensitivity has a maximum value of 3.0 × 10^−3^ K^−1^ and the relative sensor sensitivity has maximum value of 6.0 × 10^−3^ K^−1^ at 301 K. The present material also shows its sensing ability with the variation in the range of temperature from 301 K to 1173 K.

## 5. Conclusions

In conclusion, the NaGdF_4_: Er^3+^/Yb^3+^ upconversion nanoparticles synthesized via the thermal decomposition route show a nanostructure with around 20 nm particle size. The α-phase NaGdF_4_ crystal structure was confirmed by XRD analysis and particle shape/size was analyzed through FESEM analysis. The multifunctional luminescence-like upconversion and cathodoluminescence was successfully observed using 980 nm laser and electron beam excitations. The luminescence mechanisms for both luminescence processes were successfully explained on the energy level diagram. Finally, the synthesized particle was utilized for a non-contact type temperature-sensing application in a wide range of temperatures, from 300 to 1173 K, and a high absolute sensor sensitivity of 4.0 × 10^−3^ K^−1^ and relative sensor sensitivity of 6.0 × 10^−3^ K^−1^ at 301 K were observed. Overall, the present sample displays different luminescence emission properties and has potential applications in the temperature-sensing and display fields.

## Figures and Tables

**Figure 1 materials-15-06563-f001:**
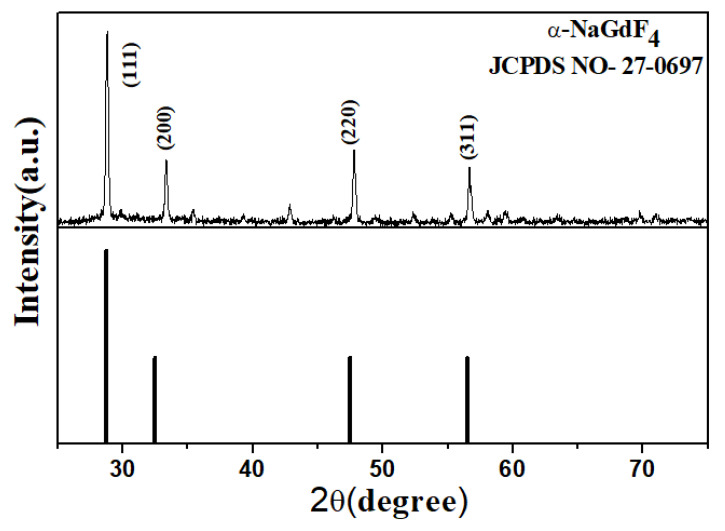
The XRD analysis of NaGdF_4_: Er^3+^/Yb^3+^ phosphor particles for α-phase formation.

**Figure 2 materials-15-06563-f002:**
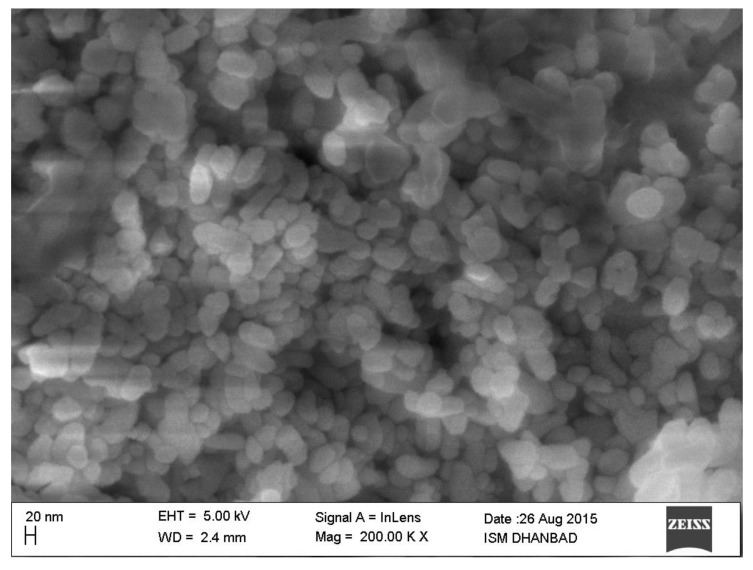
The field emission scanning electron microscopy (FE-SEM) image of NaGdF_4_: Er^3+^/Yb^3+^ phosphor particles.

**Figure 3 materials-15-06563-f003:**
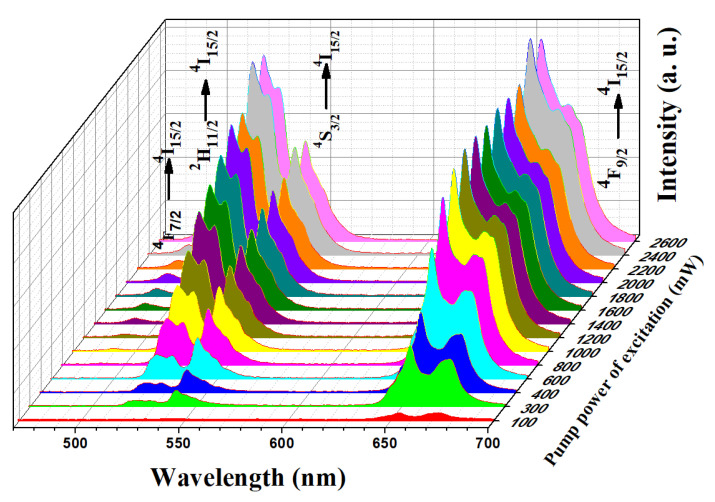
Power-dependent upconversion emission spectra analysis of NaGdF_4_: Er^3+^/Yb^3+^ at room temperature excited by a 980 nm diode laser source in the range of 100–2200 mW pump power of excitation.

**Figure 4 materials-15-06563-f004:**
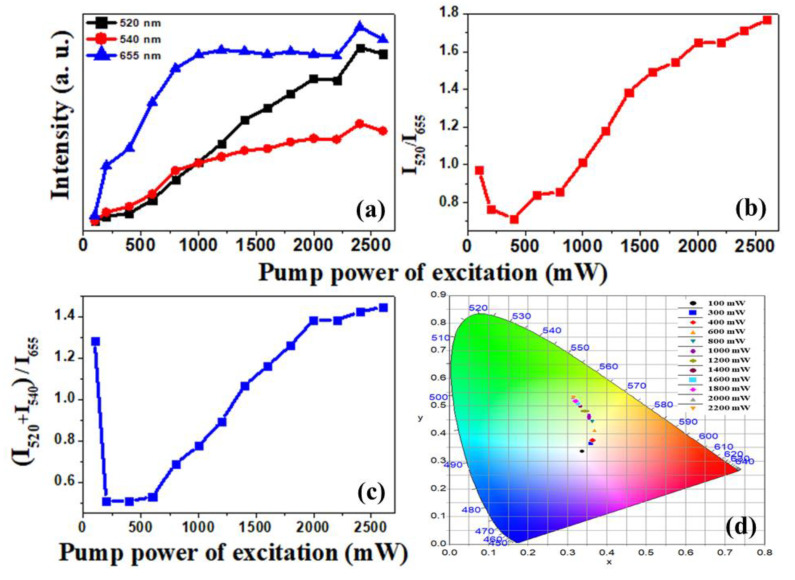
(**a**) A comparison of the variation of upconversion luminescence intensity with the pump power (from 100 mW to 2600 mW) at different emission bands of 520 nm, 540 nm, and 655 nm. (**b**) The ratio of intensities of the thermally coupled levels at 520 nm and 540 nm corresponds to each excitation power. (**c**) The variation of the ratio of total green to red intensities corresponds to the pump power of excitations. (**d**) CIE (International Commission on Illumination chromaticity diagram) color coordinates (x, y) representation on CIE color diagram of samples with different excitation powers.

**Figure 5 materials-15-06563-f005:**
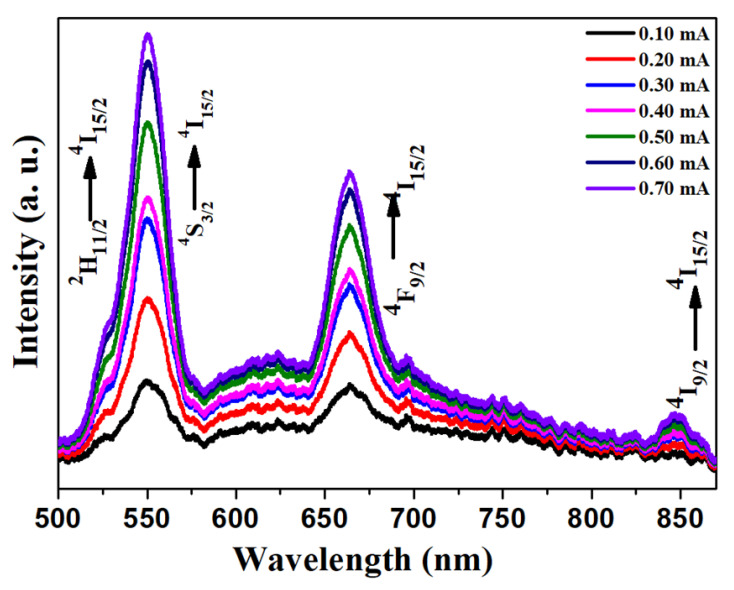
Filament current-dependent cathodoluminescence emission spectra variation of NaGdF_4_: Er^3+^/Yb^3+^ upconversion phosphor particles.

**Figure 6 materials-15-06563-f006:**
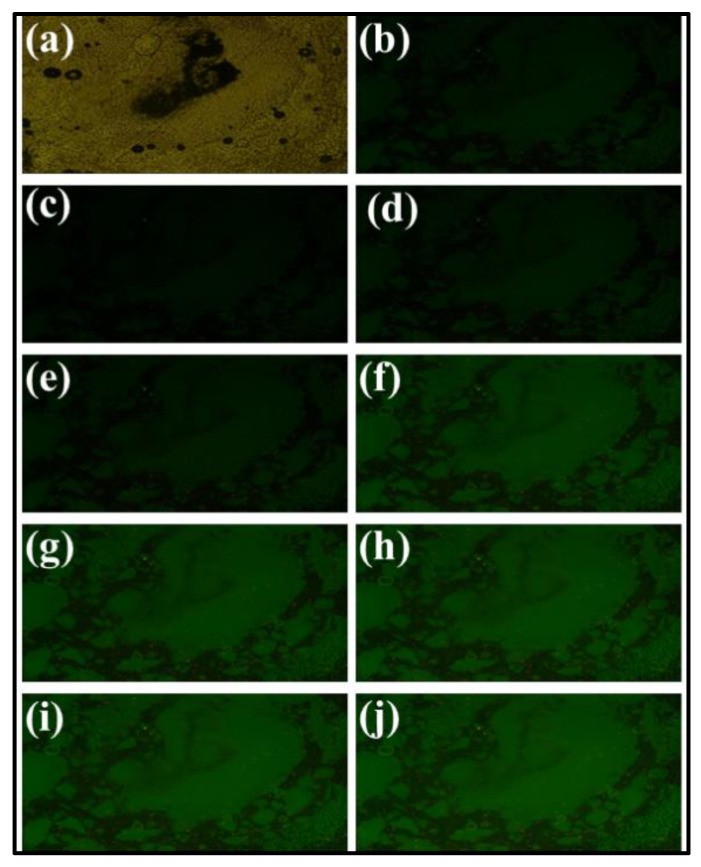
(**a**) Photograph of the sample prepared with NaGdF_4_: Er^3+^/Yb^3+^ upconversion phosphor particles for cathodoluminescence under the illumination of white light. (**b**–**j**) Photomicrograph in transmitted light at different filament currents of canal rays from 0.10 mA to 0.70 mA and the respective CL images.

**Figure 7 materials-15-06563-f007:**
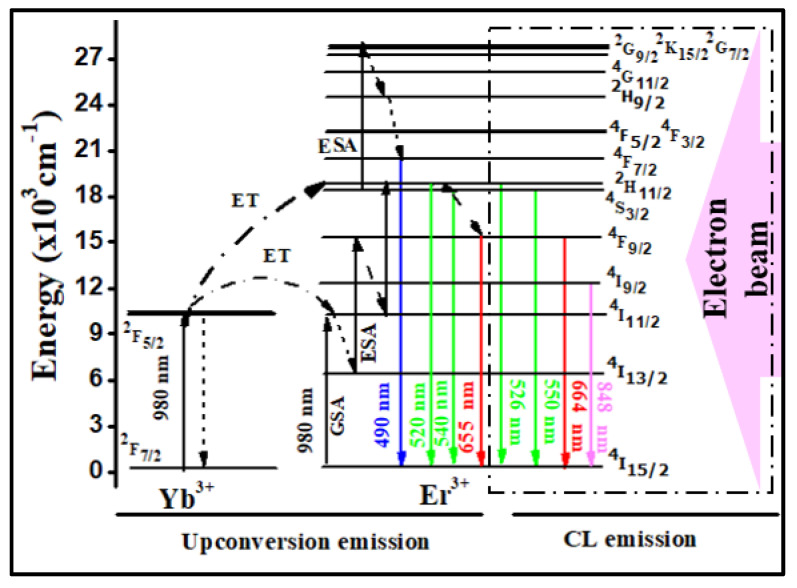
Energy level pathways for upconversion emissions bands of Yb^3+^ (sensitizer) and Er^3+^ (activator) with possible excitation for NaGdF_4_: Er^3+^/Yb^3+^ upconversion phosphor particles. Along the side, we explain the CL emissions due to the excitation of canal rays.

**Figure 8 materials-15-06563-f008:**
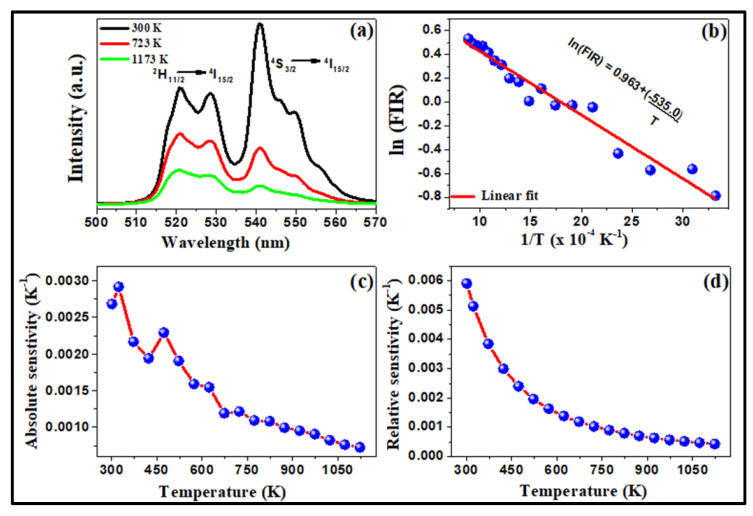
Temperature sensing behavior of NaGdF_4_: Er^3+^/Yb^3+^ upconversion phosphor particles sampled at low power excitation of 500 mW. (**a**) Samples surrounding temperature-dependent upconversion emission intensity variations at different selected temperatures; (**b**) ln (FIR) with the function of inverse temperature; (**c**) variation of absolute sensor sensitivity with temperature; (**d**) variation of relative sensor sensitivity with temperature.

**Table 1 materials-15-06563-t001:** Comparative results on temperature-sensing abilities of Er^3+^/Yb^3+^ doped/co-doped particles.

S. No	System	Temperature Range (K)	Sensor Sensitivity (K^−1^)	Reference
1	NaGd(WO_4_)_2_: Er^3+^/Yb^3+^	293–573	0.0119 K^−1^ at 453 K	[24]
2	YVO_4_: Er^3+^/Yb^3+^	300–485	0.0116 K^−1^ at 380 K	[25]
3	BaTiO_3_: Er^3+^/Yb^3+^/Zn	120–505	0.0047 K^−1^ at 430 K	[26]
4	NaYF_4_: Er^3+^/Yb^3+^	93–673	0.0029 K^−1^ at 368 K	[27]
5	NaLuF_4_: Er^3+^/Yb^3+^	303–523	0.0052 K^−1^ at 300 K	[28]
6	β-NaGdF_4_: Er^3+^/Yb^3+^	303–563	0.0037 K^−1^ at 300 K	[29]
7	NaYF_4_: Er^3+^/Yb^3+^/Gd^3+^/Nd^3+^	288–328	0.0026 K^−1^ at 300 K	[30]
8	ZnO/TeO_2_: Er^3+^/Yb^3+^	300–430	0.0120 K^−1^ at 429 K	[31]
9	NaYF_4_: Er^3+^/Yb^3+^	300–750	0.0044 K^−1^ at 637 K	[32]
11	NaGdF_4_: Er^3+^/Yb^3+^	301–1173	0.0060 K^−1^ at 301 K	Present work

## Data Availability

The data would be available on demand.

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
