# Peer review of "Upconversion Emission Studies in Er3+/Yb3+ Doped/Co-Doped NaGdF4 Phosphor Particles for Intense Cathodoluminescence and Wide Temperature-Sensing Applications"

_materials, 2022, doi:10.3390/ma15196563_

Round 1

Reviewer 1 Report

The submitted manuscript successfully synthesized the NaGdF4:Er3+/Yb3+ upconversion nanoparticle via thermal decomposition route. The multifunctional luminescence like upconversion and cathodoluminescence was successfully observed, and the luminescence mechanisms for both luminescence processes were successfully explained on the energy level diagram. However, there are still several issues that should be addressed before acceptance to any scientific journal.

1. Why choose NaGdF4 as the base material? Does the doping concentration of Er3+/Yb3+ have an impact on the upconversion performance? What is the optimal doping concentration?

2. What is the actual doping amount of Er3+/Yb3+?

3. Through the Raman spectrometer (line 126), NaGdF4 is a mixed phase of α and β, so what role does the two phases play in the upconversion performance?

4. The unit of “800 nm” in lines 147, 149 and 165 should be “mW” instead of “nm”.

5. The wavelength in line 168 uses “522 nm” and “542 nm”, while “520 nm” and “540 nm” are used in other places. Please unify.

6. UC spectra and CL spectra are different (line 215), and the explanation in the article is caused by the differences in the excitation sources. But theoretically, as long as the energy is sufficient, the electronic transition is not affected by the excitation sources. Whether more convincing evidence can be given (Such as UC spectra under excitation light of different wavelengths)?

Author Response

Hello Sir,

The reply to the first reviewer is attached herewith.

Thank you!

Reviewer 2 Report

Manuscript “Upconversion emission studies in Er3+/Yb3+ doped/codoped 2 NaGdF4 phosphor particles for intense Cathodoluminescence 3 and high-range temperature sensing applications”

This manuscript needs extensive editing of text and English. Some valuable experiments are missing (for example XRD analysis). My comments are given below:

  1. Line 26-40 Please re-write the text, there are many sentences with a similar point.
  2. Line 41 Authors wrote “impotent”. Did you mean important? Please check English.
  3. Line 43 Authors wrote” The higher upconversion emission”. It is better to use The high upconversion emission.
  4. Line 46 Table 1 should be below the text, not at the end of the manuscript!
  5. Line 46-60 Please re-write the text. There are plenty of English errors and non-sense sentences, with repetitions of the same facts.
  6. Line 83. The authors wrote, “the decomposition process is completed”. Decomposition of what?
  7. Line 84 “and the heat was re- moved while stirring”. Nonsense sentence.
  8. Please insert XRD analysis of Er3+/Yb3+ doped/codoped 2 NaGdF4 phosphor. Please in further text insert real formulae of produced material, and give the atomic percentage of dopants. (Example NAGdF4: x at.% Er; y at.% Yb). Write some sentences about crystal lattice.
  9. Figure 1 b) Please insert a Figure with higher resolution. Please insert the average diameter of produced material.
  10. The authors wrote, “Due to that 117 the FE-SEM surface is also having some non-celerity.” Please re-write and enhance your English in the manuscript!
  11. The authors used Raman to describe the alpha and beta phases of NaGdF4 and cited reference 24. This reference is wrong (it is about YF3: Er3+/Yb3+) and it is nonsense to use the Raman structure for the detection of alpha and beta phases of NaGdF4. As I previously commented, the authors must insert XRD analysis!
  12. Figure 2. Please describe peaks with suitable references, the reference 24 is wrong.
  13. Figure 4. Please insert concrete values of different filament currents for b-j.
  14. Line 208 You first wrote about cathodoluminescence, then in one part, you explain cathodoluminescence?! The order must be different.
  15. Line 208-221 Please re-write the text and check the English.
  16. Line 275 The authors wrote, “The multifunctional luminescence like upconversion and cathodoluminescence was successfully observed in the 276 present system.” Please rewrite.

Author Response

Hello Sir,

The reply to the second reviewer is attached herewith.

Thank you!

Reviewer 3 Report

In this article, the authors prepared NaGdF4-based UC nanoparticles for potential cathodoluminescence and temperature sensing applications. In general, the article can be considered for publication after a major revision. 

1) The manuscript should be revised by a native speaker. Typos such as "cooping", "impotent", "verity", "cherecrization", etc. 

2) The introduction is vague and devoid of facts. Please consider rewriting what has been done in this area, what similar materials have been used, and what needs to be improved. 

3) Experimental part. What was the total mass of RE hexachlorides used for the synthesis? Methanol was not added to the system - where is coming from? 

4) It is better to check morphology by TEM, rather than SEM. Provide EDX elemental mapping to confirm the even distribution of Yb and Er elements in nanoparticles.

5) It is preferable to analyze the sample using a quantitative method such as ICP-MS or ICP-OES to determine the true ratio of Na:Gd:Yb:Er. 

6) Provide XRD analysis of the sample! 

7) I don't observe any statistical analysis, each measured point (Figure 3a, b, c  & Figure 7b, c, d) should be retested several times and provided as mean value + SD (error bar charts). 

8) Introduction part can be improved with recent highly relevant studies such as doi: 10.1007/s11706-019-0482-z and doi: 10.1039/C4NR06944G.  

Author Response

Hello Sir,

The reply to the third reviewer is attached herewith.

Thank you!

Round 2

Reviewer 1 Report

The authors revised the manuscript according to the comments. It is now acceptable for me.

Author Response

Dear Reviewer,

Thank you very much for giving your consent regarding acceptance of our revised manuscript. 

Thank you and best regards!

Reviewer 2 Report

The authors did not follow suggestions from previous revision. There are plenty of English mistakes and mistakes in typing. The mayor problem is purity of material, the authors obtained mixture of fluorides where the dominant is alpha NaGdF4, but they also have impurities and constantly claimed it could be ignored. My comment are given bellow. I am very sorry, but I will not recommend accepting manuscript in present form.

The authors wrote at line 126 “These peaks may be ignored as it can be easily removed in base line correction.” This is wrong; clearly, the authors have impurities (evident peaks around 35 and 43 degrees). Please mark impurities and give the referent card for them.

Figure 2. The authors did not insert atomic percentages for dopants!

Authors have 2 Figure 2! Authors did not inserted better quality Figure for EDX analysis!

Author Response

Dear Sir,

Thank you very much for your valuable comments and suggestions. Here, I would like to inform you that in previous revisions we have tried to improve the content of the manuscript, especially for the English language. In the present revised manuscript, we have tried to improve the manuscript again to improve the quality of the content of the manuscript.

On the basis of the relative strength of intensity, it was claimed that the peaks are corresponding to the background. Now, the extra peaks were tried to match with the beta-phase of NaGdF4 JCPDS file number 270699. Through this matching, the results were observed that no peak of this plot was matched with the standard data file for the beta phase. Therefore, the small peaks are mostly corresponding to the background.

In the revised manuscript due to not having a good quality EDX image here we have decided to eliminate this analysis from the manuscript as we do not have enough time to recharacterize the sample for better image quality. After this revision here we are submitting this manuscript again for consideration. Please, receive this revised form of the manuscript.

Thank you and best regards!

Reviewer 3 Report

The majority of raised concerns were resolved by the author. However, one of them still need to be addressed. 

- Provide EDX elemental mapping to confirm the even distribution of Yb and Er elements in nanoparticles. This question was missed. 

Author Response

Dear Reviewer,

Thank you very much for considering our manuscript for publication in MDPI Advanced Nanomaterials and Nanotechnology,
Sir, The EDX spectrum analysis was done and given in figure 2 (b) with detailed analysis in respective sections. In this image, the presence of Er and Yb peaks is strongly supporting our claim that the Er and Yb elements are present in the sample. Another piece of evidence is the optical characterization of the present sample. The 980 nm excitation energy is confirming the presence of the Yb element in the present sample and different emission bands of upconversion luminescence are evidence of the presence of Er element [Figures 3]. In figure 7 the same thing has been explained on the energy level diagram. 
Sir, we have a very short duration of time for the revised submission of this manuscript and we do not have EDX elemental mapping facility nearby to us. For this purpose, we need sufficient time to arrange this characterization somewhere else.
 Therefore, we are requesting you that consider the above evidence in support of the Presence of Er and Yb elements in the sample and accept this manuscript for publication. 

Thank you and best regards!